# Relevant Design Aspects to Improve the Stability of Titanium Dental Implants

**DOI:** 10.3390/ma13081910

**Published:** 2020-04-18

**Authors:** M. Herrero-Climent, P. López-Jarana, B. F. Lemos, F. J. Gil, C. Falcão, J. V. Ríos-Santos, B. Ríos-Carrasco

**Affiliations:** 1Porto Dental Institute, 4150-518 Porto, Portugal; dr.herrero@herrerocliment.com (M.H.-C.); plopezjarana@gmail.com (P.L.-J.); bpvflemos@gmail.com (B.F.L.); cfalcao@ufp.edu.pt (C.F.); 2Faculty of Health Sciences, Fernando Pessoa University, 4249-004 Porto, Portugal; 3Faculty of Dentistry, Bioengineering Institute of Technology, International University of Catalonia, 08017 Barcelona, Spain; xavier.gil@uic.es; 4Department of Periodontology, University of Seville, 41009 Seville, Spain; brios@us.es

**Keywords:** dental implant, tapered implant, implant design, immediate loading, insertion torque, ISQ, RFA

## Abstract

Post-extractional implants and immediate loading protocols are becoming much more frequent in everyday clinical practice. Given the existing literature about tapered implants, the objective of this paper was to understand whether implant shape had a direct influence on the results of the insertion torque (IT) and implant stability quotient (ISQ). Seven tapered implant prototypes were developed and distributed into three groups and compared with a control cylindrical implant—VEGA by Klockner Implant System. The implants were inserted into bovine bone type III according to Lekholm and Zarb Classification. The sample size was n = 30 for the three groups. Final IT was measured with a torquemeter, and the ISQ was measured with Penguin Resonance Frequency Analysis (RFA). Modifications done to the Prototype I did not reveal higher values of the ISQ and IT when compared to VEGA. In the second group, when comparing the five prototypes (II–VI) with VEGA, it was seen that the values of the ISQ and IT were not always higher, but there were two values of the ISQ that were statistically significantly higher with the 4.0 mm diameter Prototypes II (76.3 ± 6.1) and IV (78 ± 3.7). Prototype VII was the one with higher and significant values of the ISQ and IT. In both diameters and in both variables, all differences were statistically significant enough to achieve the higher values of primary stability values (IT and ISQ). Given the limitations of this study, it can be concluded that when there is an increase of the diameter of the implant and body taper, there is an increase of the ISQ and IT, showing that the diameter of the implant is an important criteria to obtain higher values of primary stability.

## 1. Introduction

The use of dental implants is currently a common procedure in dental practice, and its use as another therapeutic tool in treatment plans for oral rehabilitation is a daily occurrence with a 95%–100% success rate [1,2,3], thus showing the high predictability of implant treatments and the simplicity of their application in most of the procedures in which they are necessary [4]. Nowadays, scenarios considered complex or risky years ago, like post-extractional implants and immediate loading, have become frequent clinical procedures [5]. The review of Slagter et al. showed that the one-year survival rate of single post-extractional implant placement in the aesthetic zone was 97.1% [6,7,8,9]. Immediately loaded and conventionally loaded implants have shown implant survival rates of 98.2% and 98.5%, respectively, after two years [10].

Most patients do not want to be toothless, which explains why the immediate loading procedure and/or post-extractional implants have become a common option in most dental practices [11].

Both surgical protocols for post-extractional implants and immediate loading depend on diverse critical factors [12]. One of these crucial factors is the primary stability of the implant when it is inserted into bone [13]. The cumulative survival rate for post-extractional dental implants and immediate loading (before seven days after implant installation), as published by Gallucci et al. in 2018 in the ITI (International Team for Implantology) Consensus, was 98.4% (median 100%; range 87.5%–100%) with a mean follow-up of 28.9 months (SD = 15.2; range 12–60). The success rates ranged from 87% to 100% [9,12,14].

As mentioned previously, primary stability is a key factor to consider on post-extractional implants and immediate loading [15]. It is a mechanical property of the implant described by the resistance forces that an implant achieves when it is inserted inside the preparation drill [16]. There are several ways to measure stability, and two of the most used are insertion torque (IT) and Resonance Frequency Analysis (RFA), both of which are used to measure the implant stability quotient (ISQ) [17]. 

Several researchers have described different techniques in order to achieve higher levels of primary stability, which is considered crucial for these procedures [18]. The macro design of the implant improves the result of primary stability and osseointegration success [19]. Different studies have evaluated an implant macro design that could be modified in order to increase the mean values of primary stability [20].

The macro design of a dental implant is made up of different features that could influence the primary stability. Therefore, the shape of the thread, the type of implant body, and even the shoulder design have been studied [21,22,23].

The morphology of threads represents another challenge to engineers because of the depth, the shape, and the thread pitch. In order to increase the implant surfaces in contact to bone and to achieve higher levels of primary stability, the threads have a long and rectangular design [24]. The thread pitch is measured by the distance between two threads on the same side of the implant [25]. It is important to take the lead of the thread pitch into consideration; this is defined as the distance between one rotation movement of the thread on an axial direction [25,26], and it can affect the insertion speed of the implant [27]. The shape of the threads is determined by their thickness and type of angle [28]. As such, we there are threads of the V, squared, reinforced, and inverse reinforcement shapes [19]. The thread pitch is the distance between the center of one thread and the next one perpendicular to the implant. The smaller the distance between them, the greater the implant surface in contact with bone, which improves the distribution of forces of it [26].

The depth of the thread is the distance from the tip of the thread to the implant body. The width of the thread is the distance between the most coronal part of the thread and the apical at the level of the tip of the thread [29].

The design of the thread must ensure a balance between the axial and non-axial forces that are generated on the implant, distributing them as well as possible to the surrounding bone [30].

The vertical sulcus around the body of the implant represents another anatomic reference point of the macro design of the implant that could help increase the mean values of their primary stability [31]. These vertical sulci could be non-self-tapping, compacting, or impacting, depending on the sense on the groove [32]. The self-tapping design has shown higher levels of stability during the osseointegration process compared to non-self-tapping implants [33].

In this sense, the implant body could be cylindrical or conical (parallel or tapered implant, respectively). Tapered implants, as compared to parallel (cylindrical) implants, are traditionally designed to provide better stability and to facilitate the surgical protocol of dental implants [34,35]. The tapered implant shows advantages on many clinical situations like low density or type III bone (Lekholm and Zarb Classification), immediate loading, and immediate implants [28,36]. Anatomic locations like the posterior with low density bone like maxilla and mandibular bones have demonstrated a 100% survival rate with tapered implant and immediate loading [34]. Recent publications on macro design implants for immediate loading show higher levels of torque insertion with tapered implants [37,38].

A clinician can measure primary stability using torque insertion values according to the manufacturer’s advisement or by registering the analysis of radiofrequency resonance (RFA) [17]. RFA is a non-invasive method to express the horizontal interface between the implant and the preparation bone drill [16]. RFA was described by Meredith in 1997 and is based on a piezoelectric system within a specific frequency meant to make the implant vibrate inside the preparation drill. Implant resistance to vibration is measured by the device and it transformed into an ISQ value (implant stability quotient within a 0–100 scale, 100 being the maximum implant stability) [16,39]. 

Insertion torque during tapping is mainly produced by the resistance forces from the cortical bone around the neck of the implant. Both methods have been used to evaluate primary stability, but these parameters could be influenced by bone equality, drill technique, and implant design [40]. The RFA could be measured by using different RFA technologies such as the Osstell ISQ instrument, which has been proven to be a repeatable and reproducible tool [16]. Recently, another device to register RFA values has become available: the Penguin RFA too. The evidence published about these new technologies showed an Intraclass Correlation Coefficient (ICC) of 0.933 and 0.944 for transducers from each system, respectively [41]. Furthermore, the in vitro studies comparing these two RFA technologies (Osstell ISQ and Penguin RFA) have demonstrated repeatability and reproducibility [42].

These RFA studies confirmed that an inverse correlation between the values of the ISQ and the lateral movement of the implant exists. This means that the ISQ is a metric that can be used to register the quantity of micromovements that can cause posterior fibro osseous integration [43,44]. In 2015, Brizuela et al. concluded that the insertion torque measures the resistance force that the preparation bone drill offers at the implant installation in an apical direction [17]. Dental implants have improved their biological, chemical, and mechanical properties to achieve better survival rates [45,46].

The aim of this article was to study the influence on primary stability mean values when several areas of the macro design of tapered implants were modified.

## 2. Materials and Methods

In the present study, all the implants used were from Klockner Implant System and were VEGA implants (bone level type implants) (SOADCO S.L., Escaldes-Engordany /Andorra).

The Klockner Implant System Company has been searching for an improvement on implant stability for their VEGA implant. For this purpose, 7 prototypes were designed and divided into 3 groups, and the results were obtained with the aim to reach a design that allowed for an increased value of the IT and ISQ. The control group was the VEGA implant, which was compared to every group. One type of bone density was used to evaluate these two variables according to the Lekholm and Zarb Classification, and this was the type III bovine kneecap. The measurements of the insertion torque were made with a Tohnichi ATG6CN torquemeter (Tohnichi Mfg. Co Ltd., Tokyo, Japan), and the measurements of the ISQ values were made with a Penguin RFA device (Integration Diagnostics, Sweden). For the ISQ values, two measurements that were perpendicular to each other and perpendicular to the MulTipeg in each implant were obtained, after which an average of the two measurements was calculated. The MulTipegs used were the following: For the 3.5 mm diameter MulTipeg, the reference was 55065, number 57, and for the 4.0 mm diameter MulTipeg, the reference was 55034, number 26.

All implants were tested in 3.5 and 4.0 mm diameters, and all were 10 mm in length. The sample size was calculated with N Query Advisor v4.0 for *p* < 0.05 based on two studies [16,47]. The calculated sample size was n = 30 for each group. 

The preparation technique was the one recommended by company (Figure 1). 

Every example of the shape and the macro design values is given with reference to the 3.5 mm diameter implants.

This study was carried out in a consecutive way; that is, as the results of the first group were obtained, the next group was made with modifications applied to it, and so on, to obtain a significant increase of the primary stability results of the ISQ and IT. 

1st Group: VEGA vs Prototype I 

The first modifications of the macro design of the VEGA implant were made in Prototype I. By maintaining the same diameter (3.55 mm) at the maximum diameter point and decreasing it in the apical diameter portion (2.5–2.0 mm), we brought an increase of the taper of the implant; there was also an increase on the thickness of the threads (0.14–0.17 mm) and the introduction of vertical grooves in a clockwise direction that made the implant self-tapping. In Figure 2, we can observe the Prototype I being compared to the VEGA.

2nd Group: Control vs Prototypes II, III, IV, V, and VI.

After getting the results from the 1st group, the 2nd group had the main objective of evaluating the difference between having vertical grooves in different numbers and directions and whether there was a significant increase of the implant stability in one of these different designs. Prototypes II and III had vertical grooves in the counter clockwise direction, which provided them with the ability to compact the bone in the apical region where they were placed. This fact produces them self-compacting implants. Prototypes IV and V, like Prototype I, had vertical grooves in the clockwise direction, so these were self-taping implants. Prototype VI was the only one to have vertical grooves in the direction of the apex. Additionally, there was an increase of 0.1 mm in the maximum diameter of all 5 Prototypes when compared to the control. The apical diameter decreased 0.6 mm from the control. These modifications had the goal to increase taper.

When compared to Prototype I, the main differences were the increase of 0.1 mm of maximum diameter, the decrease of the apical diameter of 0.1 mm, and the decrease of the thread of by 0.02 mm.

Prototype II: 3 grooves in counter clockwise direction—self-compacting. 

Prototype III: 4 grooves in counter clockwise direction—self-compacting.

Prototype IV: 3 grooves in clockwise direction—self-tapping.

Prototype V: 4 grooves in clockwise direction—self-tapping.

Prototype VI: 4 vertical grooves.

The different prototypes can be observed in Figure 3. 

3rd Group: Control Versus Prototype VII. 

After getting the results from the 2nd group, there was a necessity to have higher ISQ and IT values, so we followed these modifications for the Prototype VII:

We introduced a 0.2 mm wider maximum diameter core that was cylindrical until the last two millimeters and then conical shaped to the control implant (VEGA). The apical diameter was decreased 0.6 mm with the goal of increasing the taper. The threads also had similar tapers to the body, and there was an increase of 0.2 mm on the thread width (Figure 4). When compared with Prototypes II, III, IV, V, and VI, the main differences were the increase of 0.1 mm on the maximum diameter, the increase of 0.1 mm of the apical diameter, the last 2 mm of the core were cylindrical, and the rest of the core was conical, thus making the implant have a higher taper compared to the other 5 prototypes (+6°). There was also an increase of 0.1 mm of the thread width.

### Statistical Analysis

To determine if there were statistically significant differences between the different studied variables, the Minitab 16 Statistical Software was used. If the values met a normal distribution (*p* > 0.05) and 2 groups of independent data could be compared, the statistical analysis was performed using the Student’s parametric *t*-test. When the values did not meet a normal distribution (*p* < 0.05) and two groups of independent data were compared, the analysis was performed with a non-parametric Mann–Whitney test.

## 3. Results

1st Group: Control vs Prototype I

Prototype I was shown to have higher ISQ values, but they were not statistically significant.

With the modifications that were made, the increase that was made in the thread thickness, the reduction of apical diameter, and the presence of the longitudinal sulcus in the control implant were not enough to have a significant increase of the IT and ISQ.

2nd Group: Control vs Prototypes II, III, IV, V, and VI.

The five new prototypes did now show significant differences of IT and ISQ values at the 3.5 mm diameter.

In the 4.0 mm diameter implants, there was a significant difference of the ISQ, mainly with Prototypes II (76.3 ± 6.1) and IV (78.0 ± 3.7) when compared to control (73.8 ± 6.8).

When comparing them with each other, none of the five studied prototypes showed a better performance of the IT or ISQ. From these results, it was seen that none of the longitudinally-shaped sulci offered better ISQ or IT values.

3rd Group: Control vs Prototype VII

There was an increase of the IT and ISQ values when compared to the control. Prototype VII was shown to have significant differences in IT and ISQ values in type III bone with both 3.5 and 4.0 mm diameter implants. (*p* < 0.05).

## 4. Discussion

Primary stability is a crucial factor for the process of osseointegration. Besides that, it is an important factor in certain protocols like post-extractional implants and even more so in immediate loading. For these protocols, it is essential that stability levels are as high as possible [28].

Conical-designed implants have been shown to achieve better levels of primary stability than those with parallel walls. Even in situations that are not very favorable to the protocols described above, such as poor bone quality, conical implants have obtained better levels of stability. In this sense, the authors of this paper wanted to review the effect on stability levels by varying different parts of the macroscopic design of the implant.

According to Ryu et al. 2014 in a literature review, it has been concluded that if the main goal is primary stability, then square-shaped threads are the ones that offer the higher levels of the IT and ISQ [26].

Taking this into account, all the prototypes created here had squared-shaped threads.

In group 1 of the ITI consensus in 2018, Jung et al. focused on the influence of length and implant design by analyzing a total of 29 articles, including three RCTS with three years of follow-up. In total, 245 patients and 388 implants were studied. Clinical recommendations specify that tapered implants may be an option to consider in cases of damage to anatomical structures or case at risk of producing some apical fenestration. Regarding the use of these implants as tools to achieve better levels of insertion torque, the authors considered tapered implants as an alternative, although the long-term results are still not clear [48]. 

For the 3.5 mm diameter Prototype I, there was not an increase of IT compared to VEGA (IT values: VEGA = 30.1 Ncm/Prototype I = 22.2 Ncm/*p* = 0.025). (Table 1). On the other hand, there was an increase of the ISQ, even though it was not statistically significant (ISQ values: VEGA = 71.4/Prototype I = 71.5/*p* = 0.6841). For the 4.0 mm diameter implant, there was an increase of both the IT and ISQ (IT values: VEGA-17.9Ncm/Prototype I-21.3Ncm/*p* = 0.0695) (ISQ values: VEGA = 64.8/Prototype I = 66.6/*p* = 0.454). With these findings, we could conclude that just varying the apical diameter, the shape of the body, and the presence of the lateral groove was not enough to ensure that there was a statistically significant increase for both diameters (3.5 and 4 mm) of the ISQ and IT in relation to the control implant. This led the study to continue varying implant design in order to achieve higher values of the ISQ and IT. For that, a second group was created with five new prototypes in order to evaluate if there was any difference when applying different types of vertical grooves.

In the second group) (Table 2), the comparison between Prototypes II, III, IV, V, and VI with different types of vertical grooves—either in number or in direction—did not demonstrate that there was a type of groove with better results of the ISQ or IT that were statistically significant. Only with Prototypes II and IV were there statistically significant increases of the ISQ when compared to the control (ISQ values: VEGA = 73.8/Prototype II = 76.3/Prototype IV = 78.0). At this time, we could not conclude that there was a clear best prototype design between the five regarding ISQ and IT values. Even so, the chosen type of vertical grooves were those of Prototype III because they were self-bone compacting and because of the clinical perception of being an easier implant to manage/handle by operators. 

Furthermore, there was no significant increase of ISQ and IT values, which this led the study to create another prototype.

In the comparison was made between the control and Prototype VII (which, in addition to the introduction of a smaller apical diameter (−0.5 mm), had an increase of the width of the threads (+0.02 mm) and a self-compacting longitudinal groove), there was a 0.2 mm increase in the maximum diameter, contributing to an increase of the taper of the implant body and therefore showing a statistically significant increase in the ISQ and TI for both diameters. The values obtained in Prototype VII showed that the changes in the macro design of the control implant were done according to the objectives.

Conical implants showed better results in the stability values of both the ISQ and IT in a study by Romanos et al. in 2012. The mean Periotest values (PVs) were −4.67 (±1.18) for bone level (BL), −6.07 (±0.94) for standard plus (SP), and −6.57 (±0.57) for tapered effect (TE). The mean ISQ values were 75.02 (±3.65), 75.98 (±3.00), and 79.83 (±1.85), respectively. The one-way ANOVA showed significant differences in the PVs of the three implant designs (*p* < 0.0001) and for the ISQ between the BL/TE or SP/TE implants (*p* < 0.0001) [49]. Romanos et al. 2012 showed that the use of conical implants could bring an increased implant stability. Similar results were found in the present study when comparing the ISQ values from VEGA (3.5 mm diameter = 74.9/4.0 mm diameter = 76.0) with Prototype VII (3.5 mm diameter = 78.2/4.0 mm diameter = 78.5), both with *p* < 0.005. (Table 3.)

Irinakis et al. 2009 found the following mean values results with NobelActive for insertion torque: for 43 implants of 3.5 mm in width, they found a mean value of 44 Ncm; for the implants of 4.3 mm in width, they found a mean value of 56.2 Ncm; and for the implants placed in bone soft and medium, they found a mean value of 47.9 Ncm [34]. The results described in that study were similar to what was found in the present study, as when we increased the widest platform of the implant, we increased the IT values; see Prototype VII’s IT values (3.5 mm diameter = 54.2 Ncm/4.0 mm diameter = 64.7 Ncm).

In 2015, Yamagushi et al. studied nine implant designs: a total of 90 implants (Straumann: standard RN, bone level RC, tapered effect RN; and Nobel Biocare: Brånemark MKIII, MKIV) were placed in type IV artificial bones. The torque-time curves were distributed into initial, parallel, tapered, and platform areas. The mean torque rise rate of the parallel area was smallest at 0.36 N · cm/s, with a significant difference from those of the other areas (*p* < 0.05). Values of 2.14, 2.33, and 2.65 N · cm/s were obtained for the initial, tapered, and platform areas, respectively. The results displayed that increasing the implant taper angle appeared to increase the torque rate. The researchers found that torque was mainly generated from the tapered effect because of the bone-condensing design [50]. According to the results of the present study (Table 3), it was found that the tapered implant, Prototype VII (3.5 mm diameter = 54.2 Ncm/4.0 mm diameter = 64.7 Ncm), generated more IT than the control implant, VEGA (3.5 mm diameter = 34.6 Ncm/4.0 mm diameter = 43.6 Ncm).

Karl et al. 2017 compared three different shaped implants (Astra, Dentsply—cylindrical; BLT, Straumann—tapered; NA, Nobel—tapered) and their primary stability, ISQ, and IT on a polyurethane foam with varying densities. The NA implant was the one with highest IT value of 36.52 Ncm and the one with highest values of the ISQ—53.9. This study concluded that tapered implants showed higher values of the ISQ and IT when compared to cylindrical ones [21].

The results of the third group were in agreement with the conclusions of the previous report (Karl et al. 2017) [21], which means that Prototype VII had an increased taper, from 10° to 16°, and may have contributed to an increase of the IT values (3.5 mm diameter IT: VEGA-34.6 Ncm/Prototype VII-54.2 Ncm/*p* < 0.005) (4.0 mm diameter IT: VEGA-43.6/Prototype VII-64.7/*p* < 0.005).

Like most in vitro studies, this one has some limitations regarding variability of the bovine bone and bone density in every piece, which are factors that probably explain the standard deviation found in the results.

## 5. Conclusions

Given the limitations of this study, it can be concluded that modifying the implant body shape from conical to an increased tapered angle by only reducing the apical diameter and the introduction of a self-taper sulcus was not enough to increase primary stability. For that reason, when there was an increase on the maximum diameter of the implant and, consequently, an increase of the body taper, it was clear that there was an increase of the ISQ and IT, showing that the maximum diameter of an implant is an important criteria for obtaining higher values of primary stability.

## Figures and Tables

**Figure 1 materials-13-01910-f001:**
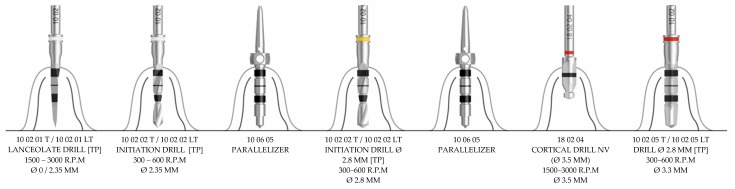
Preparation technique for the 3.5 mm diameter implants.

**Figure 2 materials-13-01910-f002:**
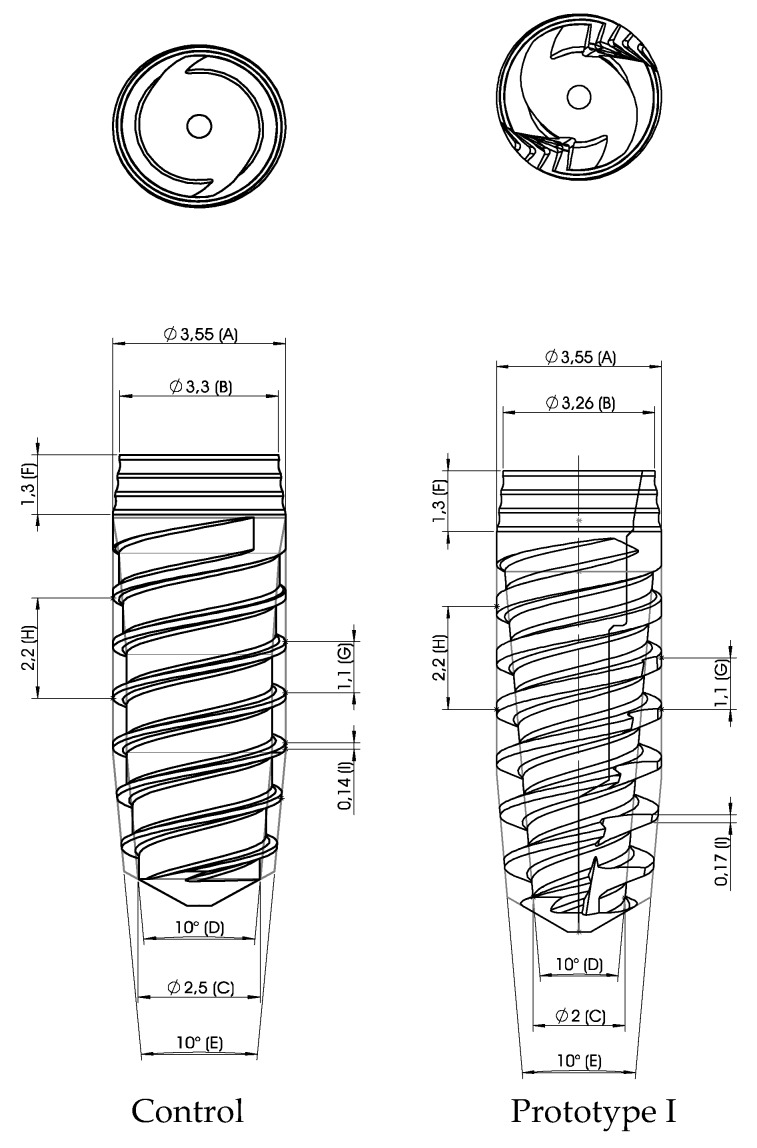
Control design and Prototype I with a more conical core, increased thickness of threads, and the introduction of helicoidal grooves.

**Figure 3 materials-13-01910-f003:**
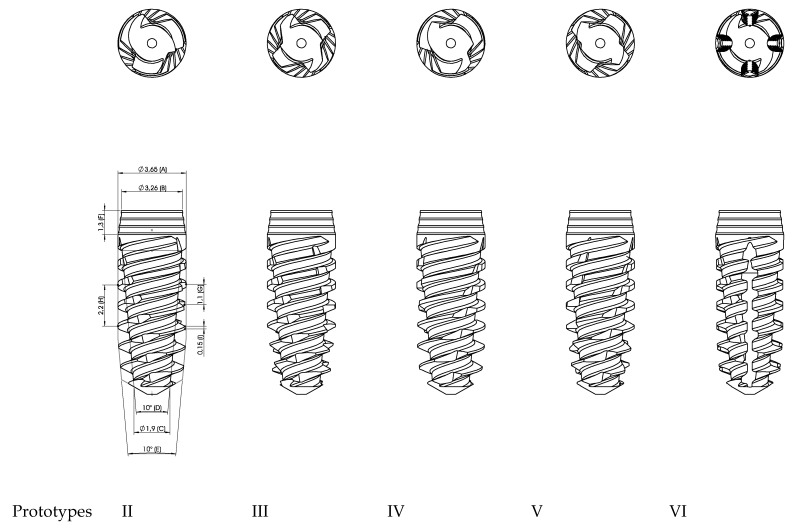
Prototypes II, III, IV, V, and VI with different vertical grooves in different numbers and directions.

**Figure 4 materials-13-01910-f004:**
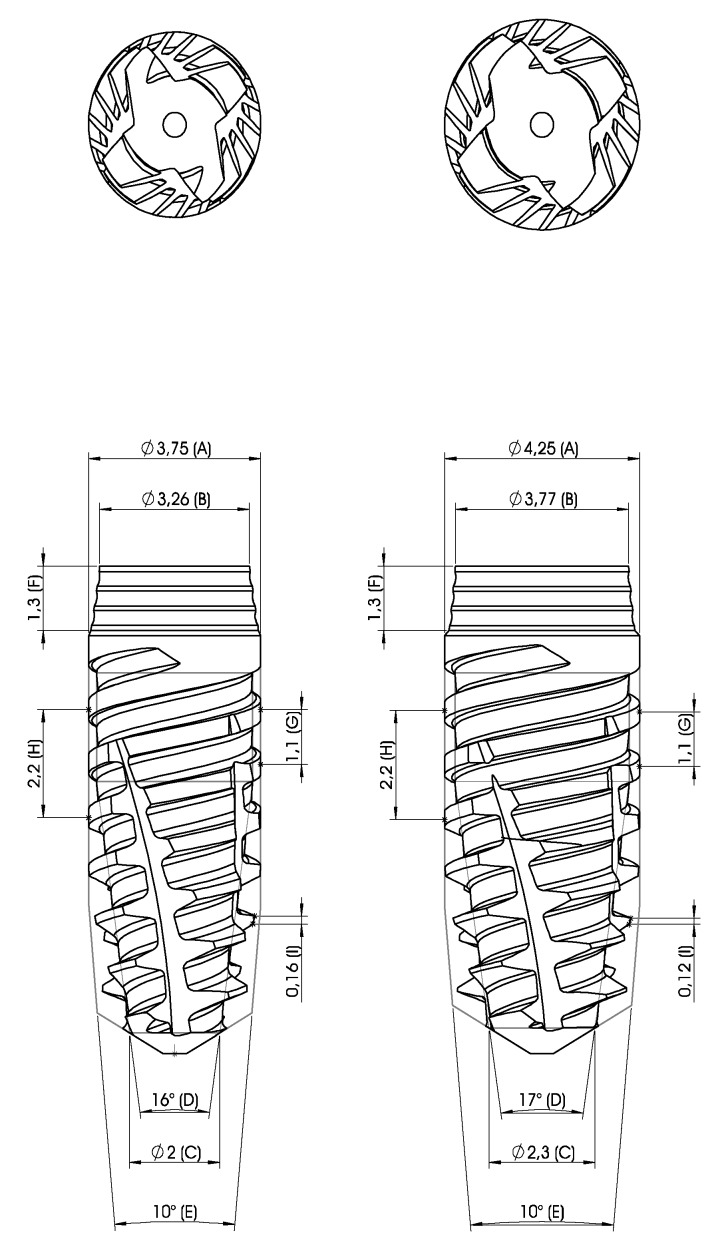
Prototype VII with two diameters.

**Table 1 materials-13-01910-t001:** Values of the insertion torque (IT) and implant stability quotient (ISQ) from the 1st group, the control, and Prototype I with 3.5 and 4.0 mm diameters; the yellow values are the statistically significant ones.

Implant	IT – N/cm			ISQ		
	Mean	*p* Value	SD	Mean	*p* Value	SD
VEGA—3.5	30.1	0.025	16.3	71.4	0.6841	7.8
Prototype I—3.5	22.2		8.2	71.5		6.1
VEGA—4.0	17.9	0.0695	10.8	64.8	0.454	10.7
Prototype I—4.0	21.3		11.2	66.6		7.9

**Table 2 materials-13-01910-t002:** Values of the IT and ISQ from the 2nd group—control and Prototypes II, III, IV, V, and VI with 3.5 and 4.0 mm diameters; the yellow values are the statistically significant ones.

Implant	IT – N/cm			ISQ		
	Mean	*p* Value	SD	Mean	*p* Value	SD
VEGA—3.5	28.7	-	14.9	71.9	-	8.9
Prototype II—3.5	27.9	0.267	13.5	75.4	0.051	4.9
Prototype III—3.5	26.7		16.9	74.6		4.7
Prototype IV—3.5	25.8		14.2	74.8		5.5
Prototype V—3.5	23.2		11.8	72.1		6.9
Prototype VI—3.5	22.3		12.7	71.4		7.8
VEGA—4.0	37.8	-	20.4	73.8	-	6.8
Prototype II—4.0	36.7	0.625	19.7	76.3	0.012	6.1
Prototype III—4.0	30.2		13.5	75.0		7.7
Prototype IV—4.0	37.8		17.9	78.0		3.7
Prototype V—4.0	35.6		15.7	75.7		5.4
Prototype VI—4.0	34.4		17.8	75.5		6.2

**Table 3 materials-13-01910-t003:** Values of the IT and ISQ from the 3rd group—control and Prototype VII with 3.5 and 4.0 mm diameters; the yellow values are the statistically significant ones.

Implant	IT – N/cm			ISQ		
	Mean	*p* Value	SD	Mean	*p* Value	SD
VEGA—3.5	34.6	0.000	9.9	74.9	0.0001	5.3
Prototype VII—3.5	54.2		22.6	78.2		7.5
VEGA—4.0	43.6	0.0004	25.5	76.0	0.0192	5.0
Prototype VII—4.0	64.7		22.8	78.5		3.2

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
