# Peer review of "Relevant Design Aspects to Improve the Stability of Titanium Dental Implants"

_materials, 2020, doi:10.3390/ma13081910_

Round 1
Reviewer 1 Report
The manuscript can be of interest and considered for publication after the above revisions are made.

Author Response
Response to the 1st Referee
Dear Referee,
Thank you very much for your comments, as they were extremely beneficial to the further development of this article. In that sense, we considered all your observations and suggestions, and made the necessary changes in the article. In the following lines, we will present you with the adjustments that were made.
- “Lines 38-109 – there are no references or citations regarding each sentence. Please add references to every sentence in the introduction. The authors refer to “immediate implant” – what does immediate implant actually mean? I assume they mean post-extractional implant, therefore I recommend to refer to post-extractional implants rather than “immediate implants”. This would be more explicit for the reader.”
As your recommendation the referees references have been inserted in all sentences of the introduction in order to be more explicit. Throughout the text, the term "immediate implant" was replaced by "post-extraction implant”.
- “Line- 87-91 – please be more explicit, rephrase according to English grammar and language.
Overall poor English in the Introduction. I recommend a professional English language evaluation and rewriting.”
The English has been reviewed and corrected.
- “Discussion
there are no references or citations regarding each sentence. Please add references to every sentence in the discussion section.
Overall poor English in the discussion section. I recommend a professional English language evaluation and rewriting.”
All the citations and references have been included, and once again the English has been reviewed and corrected.
We appreciate your attention,
Best regards.
Reviewer 2 Report
The paper seems interesting to me. However, there are few concerns from my side
- The novelty of the present study is not well shown!!
- the abstract is too long and must cut down to half
- the experimental methods are not much clear as the reader really confused between the proposed groups.
- it needs to further elaborate about the obtained results
- change the comma to period when you write down any number in the table (30,1 should be 30.1)
- it is unclear how the smaller diameter has a significant effect on IT and ISQ. it is not clear yet.
Author Response
Response to the 2nd Referee
Dear Referee,
Thank you very much for your comments, as they were extremely beneficial to the further development of this article. In that sense, we considered all your observations and suggestions, and made the necessary changes in the article. In the following lines, we will present you with the adjustments that were made.
- “The novelty of the present study is not well shown!!
- “The abstract is too long and must cut down to half”
The text was fully revised and modified to become clearer and easier to interpret. The abstract was also shortened to 265 words.
3 "The experimental methods are not much clear as the reader really confused between the proposed groups.”
4 “It needs to further elaborate about the obtained results”
5 “Change the comma to period when you write down any number in the table (30,1 should be 30.1)”
All materials and methods were revised, and we attempted to create a logical sequence of the study. To achieve this, a logical timeline of the prototypes was incorporated in the study, showing and justifying the modifications made in each step to reach higher implant stability.
To accomplish the intended results regarding implant stability, groups were tested in a consecutive manner. Considering the results of each group, the needed modifications would be carried out to improve the following group. This process was repeated in 3 consecutive times.
Each decimal comma was changed to a period.
6 “It is unclear how the smaller diameter has a significant effect on IT and ISQ. it is not clear yet.”
The concept of reducing the diameter of the implant's apex and increasing the maximum diameter point aimed to modify the implant's design by increasing its taper morphology, thus comparing different conicities and their effect on primary stability.
We appreciate your attention,
Best regards.
Reviewer 3 Report
1. The language of this manuscript needs extensive improvement.
2. "7 tapered implants prototypes were developed and distributed into 3 groups and always compared with a control cylindrical implant, VEGA by Klockner Implant System."
Klochner Implant Systems Company provides seven prototype tapered implants (I - VII) to be compared with their standard cylindrical implant, VEGA, for effects on IT and ISQ. The comparisons were done in three separate experiments. In one experiment VEGA and prototype I were compared. In a second experiment, VEGA and prototypes II - VI were compared, and in a third experiment VEGA and prototype VII were compared.
Presumably, there was some logistical reason for not comparing all implants in one experiment. This should be explained in the Materials and methods, or commented on in the Discussion.
3. The abstract refers to using bovine kneecaps. Materials and methods states that two bone types were used -- kneecaps and ribs. The results in Table 1 would presumably represent one bone type without specifying which bone type was used. Please clarify this.
Author Response
Response to the 3rd Referee
Dear Referee,
Thank you very much for your comments, as they were extremely beneficial to the further development of this article. In that sense, we considered all your observations and suggestions, and made the necessary changes in the article. In the following lines, we will present you with the adjustments that were made.
- “The language of this manuscript needs extensive improvement.”
The English has been reviewed and corrected.
- “"7 tapered implants prototypes were developed and distributed into 3 groups and always compared with a control cylindrical implant, VEGA by Klockner Implant System."
Actually Klockner Implant Systems Company provides seven prototype tapered implants (I - VII) to be compared with their standard cylindrical implant, VEGA, for effects on IT and ISQ. The comparisons were done in three separate experiments. In one experiment VEGA and prototype I were compared. In a second experiment, VEGA and prototypes II - VI were compared, and in a third experiment VEGA and prototype VII were compared.
Presumably, there was some logistical reason for not comparing all implants in one experiment. This should be explained in the Materials and methods, or commented on in the Discussion.”
Material and methods were revised and it was explained the development of the study.
All materials and methods were revised, and we attempted to create a logical sequence of the study. To achieve this, a logical timeline of the prototypes was incorporated in the study, showing and justifying the modifications made in each step to reach higher implant stability.
To accomplish the intended results regarding implant stability, groups were tested in a consecutive manner. Considering the results of each group, the needed modifications would be carried out to improve the following group. This process was repeated in 3 consecutive times.
Each decimal comma was changed to a period.
- “The abstract refers to using bovine kneecaps. Materials and methods states that two bone types were used -- kneecaps and ribs. The results in Table 1 would presumably represent one bone type without specifying which bone type was used. Please clarify this.”
The study was done only on bovine kneecap, according to the classification of bone density from Lekholm and Zarb, type III.
We appreciate your attention,
Best regards.
Round 2
Reviewer 2 Report
the manuscript has been modified and it seems suitable for publication. However, I advise sharing this article with another potential reviewer.